# Metabolomic study of marine *Streptomyces* sp.: Secondary metabolites and the production of potential anticancer compounds

**Marcelo M. P. Tangerina**[1]*, **Luciana Costa Furtado**[2], **Vida M. B. Leite**[3], **Anelize Bauermeister**[2,4], **Karen Velasco-Alzate**[2], **Paula C. Jimenez**[5], **Leandro M. Garrido**[3], **Gabriel Padilla**[3], **Norberto P. Lopes**[4], **Leticia V. Costa-Lotufo**[2], **Marcelo J. Pena Ferreira**[1]*

1 Departamento de Botânica, Instituto de Biociências, Universidade de São Paulo, São Paulo, São Paulo, Brazil, 2 Departamento de Farmacologia, Instituto de Ciências Biomédicas, Universidade de São Paulo, São Paulo, São Paulo, Brazil, 3 Departamento de Microbiologia, Instituto de Ciências Biomédicas, Universidade de São Paulo, São Paulo, São Paulo, Brazil, 4 NPPNS, Departamento de Ciências Biomoleculares, Faculdade de Ciências Farmacêuticas de Ribeirão Preto, Universidade de São Paulo, Ribeirão Preto, São Paulo, Brazil, 5 Departamento de Ciências do Mar, Instituto do Mar, Universidade Federal de São Paulo, Santos, São Paulo, Brazil

* marcelotangerina@usp.br (MMPT); marcelopena@ib.usp.br (MJPF)

**Data Availability Statement:** All relevant data are within the paper and its Supporting Information files.

## Abstract

Resorting to a *One Strain Many Compounds* (OSMAC) approach, the marine *Streptomyces* sp. BRB081 strain was grown in six different media settings over 1, 2, 3 or 7 days. Extractions of mycelium and broth were conducted separately for each media and cultivation period by sonication using methanol/acetone 1:1 and agitation with ethyl acetate, respectively. All methanol/acetone and ethyl acetate crude extracts were analysed by HPLC-MS/MS and data treatment was performed through GNPS platform using MZmine 2 software. In parallel, the genome was sequenced, assembled and mined to search for biosynthetic gene clusters (BGC) of secondary metabolites using the AntiSMASH 5.0 software. Spectral library search tool allowed the annotation of desferrioxamines, fatty acid amides, diketopiperazines, xanthurenic acid and, remarkably, the cyclic octapeptides surugamides. Molecular network analysis allowed the observation of the surugamides cluster, where surugamide A and the protonated molecule corresponding to the B-E isomers, as well as two potentially new analogues, were detected. Data treatment through MZmine 2 software allowed to distinguish that the largest amount of surugamides was obtained by cultivating BRB081 in SCB medium during 7 days and extraction of culture broth. Using the same data treatment, a chemical barcode was created for easy visualization and comparison of the metabolites produced overtime in all media. By genome mining of BRB081 four regions of biosynthetic gene clusters of secondary metabolites were detected supporting the metabolic data. Cytotoxic evaluation of all crude extracts using MTT assay revealed the highest bioactivity was also observed for extracts obtained in the optimal conditions as those for surugamides production, suggesting these to be the main active compounds herein. This method allowed the

**Funding:** This research was supported by São Paulo Research Foundation (FAPESP) [grants 2017/16606-6 (M.M.P.T.), 2017/18235-5 (L.C.F.), 2017/17648-4 (A.B.) and 2015/17177-6 (L.V.C.L.)] and the Coordination for the Improvement of Higher Education Personnel - CAPES [Finance Code 001, Brazil]. L.V.C.L., M.J.P.F and N.P.L. were funded by a fellowship from the Brazilian National Council for Scientific and Technological Development - CNPq. This study is registered with the Sistema Nacional de Gestão do Patrimônio Genético e do Conhecimento Tradicional Associado [SisGen # A0031D5] within the Brazilian Environmental Ministry, and has been authorized by Instituto Florestal, within the São Paulo State Environmental Secretary [process # 260108 – 004.258/2018]. The funders had no role in study design, data collection and analysis, decision to publish, or preparation of the manuscript.

**Competing interests:** The authors have declared that no competing interests exist.

identification of compounds in the crude extracts and guided the selection of best conditions for production of bioactive compounds.

## Introduction

The oceans cover 70% of the surface of the Earth and harbors a large portion of the planet's biodiversity [1], which is further connected to a great molecular diversity of natural products found in animals, algae and microorganisms [2]. Such environment presents extreme conditions, *e.g.* high pressure, high salinity, changes in temperature, limited nutrient availability [3], among others, thus veering marine organisms to adapt to such conditions by developing unusual and, therefore, interesting metabolic pathways, which may provide complex chemical structures of relevance for biotechnological and pharmaceutical industries.

Among all the living sea creatures, microorganisms stand out for their capacity to thrive in several marine environments, from the water surface to the lower and abyssal depths; from coastal to offshore regions; from open waters to coral reefs [4]. Such a large geographical distribution throughout the oceans attests for the adaptability skills of microorganisms, merited by their genetic plasticity and rapid replication. These facts make microbes the most numerous, diverse and adaptable organisms on Earth [5]. In general, microorganisms have drawn the attention of scientists for many years due to their importance in many life processes, especially in the production of useful compounds such as vitamins, antibiotics and other pharmaceuticals [6]. In fact, nearly 70% of small molecules that are utilized as medicines are derived or inspired in natural products produced by bacteria, more specifically filamentous actinobacteria [7].

*Streptomyces* is the largest genus of the Actinobacteria phylum and the most important for the pharmaceutical industry [8]. Members of this group of Gram positive bacteria are well known for their outstanding capacity to produce active secondary metabolites, including antibiotic, antitumor, antimalarial and immunosuppressive agents [9–13]. Such variety of biological properties is due to the large diversity of structurally distinct compounds produced by over 900 species of *Streptomyces* [14]. The genus is specially known as a prolific source of antibiotics, being responsible for the production of more the half of all known antibiotics [9], with compounds already in clinical use like tetracyclines [15], chloramphenicol [16] and streptomycin [10]. Moreover, antitumor antibiotics, such as the anthracycline doxorubicin and the glycopeptide bleomycin, are also *Streptomyces*-derived compounds [17, 18]. These drugs currently in clinical use confirm the importance of the search for new therapeutic agents produced by strains from this genus.

Several parameters should be considered when conducting investigations on the production of secondary metabolites by one strain. For example, media composition, pH, temperature, oxygen availability and light intensity may affect microbes metabolism, therefore affecting compounds production [19]. These parameters can be evaluated using the *OSMAC* (One Strain–Many Compounds) approach, where a range of culture conditions are tested aiming at generating different metabolites [20, 21]. Using this approach, researchers were able to isolate from a single strain more than 20 different metabolites in yields up to 2.6 g L$^{-1}$ [22]. Such results are possible because OSMAC may turn on silent or cryptic biosynthetic genes [19], which increases the possibility of finding compounds with new structures.

One of the major obstacles nowadays in natural products research is the rediscovery of known compounds, and rapid annotation of compounds prior to isolation is a crucial and important step. In this context, metabolomics based on mass spectrometry (MS) has been

widely applied in such kind of investigations due to the high sensitivity and flexibility of this technique. MS can be coupled to liquid chromatography, which allows the analysis of minor compounds present in a sample, besides providing important structure information to help in the identification. However, MS may generate massive amounts of spectra, which makes the analysis more complex. Considering that, the online platform Global Natural Products Social Molecular Networking (GNPS) allows the use of several different MS-based metabolomics tools to analyze large sets of data. Moreover, GNPS hosts a public spectral library and many of its tools are able to automatically search for a spectral match. Molecular networking (GNPS), for instance, has been applied in screening processes for strain prioritization from large microbial libraries [23, 24], in the differentiation of intra-clade bacteria [25], for investigation of microbial communication with co-cultivation experiments [26] and along with multivariate statistical analyses in studies of culture condition optimization for the production of specific metabolites, including OSMAC approach [3, 27].

Therefore, the main goal of this study was to assess, through LC-MS/MS, the variation in the secondary metabolite production by the marine *Streptomyces* sp. BRB081 when submitted to different culture conditions. The strain was selected based on cytotoxicity assays and on the preliminary LC-MS/MS fingerprinting, which revealed the production of surugamides, a group of scarcely known octapeptides with cathepsin B inhibiting proprieties [28]. In such scenario, the main questions addressed herein were the following: (1) What is the best cultivation media for production of these compounds?; (2) How many cultivation days are needed to produce these metabolites?; and (3) Which extraction matrix, between mycelium or culture broth, provides better yields of the target compounds? Moreover, changes in total metabolite diversity from different cultivation conditions and cytotoxicity of the extracts obtained were evaluated.

## Material and methods

### Bacteria isolation and identification

Bacterial strain was recovered from the sediments collected at a depth of 0.5 m using a van Veen grab sampler at Araçá Beach, São Sebastião, SP, Brazil (23˚48'53.83"S; 45˚24'26.83"O). The sediment sample was diluted in sterile seawater, heated to 55 ˚C for 10 min and streaked onto a semi-solid nutritious medium (A1), composed of 1% soluble starch, 0.4% yeast extract, 0.4% peptone and 1.8% bacteriological agar made up with reconstituted sea water. To reduce fungal contamination, cycloheximide (10 mg L$^{-1}$) was added. Access to the collection site for this study was approved and registered with the Sistema Nacional de Gestão do Patrimônio Genético e do Conhecimento Tradicional Associado [SisGen # A0031D5] and the Sistema de Autorização e Informação em Biodiversidade [SISBIO # 49951–4], both within the Brazilian Environmental Ministry. It has also been authorized by Instituto Florestal, within the São Paulo State Environmental Secretary [process # 260108–004.258/2018].

The bacteria grew between 12 and 90 days between 26–28˚C and the phenotypic differences of the colonies including color, brightness, shape, texture, among others were the main characteristic considered to isolate them by transferring to a new dish with fresh media. The isolated strain was cultured on A1 semi-solid medium and preserved in growth broth supplemented with 50% glycerol in reconstituted sea water, aliquoted in cryogenic flasks and frozen at -80˚C.

The strain, codified by BRB081, was identified as *Streptomyces* sp. based on 16S rRNA gene sequencing (GenBank accession number JACVQE010000000) followed by comparison to the sequences present in the Ezbiocloud database for type species and also at NCBI (http://www.ncbi.nlm.nih.gov/) using Basic Local Alignment Search Tool (BLAST). The methodology used was previously detailed described by Velasco-Alzate et al., 2019 [29].

## Bacteria cultivation

Strain BRB081 was cultivated for 1, 2, 3 and 7 days in Erlenmeyer flasks (125 mL) containing 40 mL of liquid media at 28˚C and 180 rpm. Six different liquid media were used: (1) *Starch Casein Broth–SCB* (per L): soluble starch, 10.0 g; NaCl, 2.0 g; casein, 0.3 g; $KNO_3$, 2.0 g; $K_2HPO_4$, 2.0 g; $MgSO_4 \cdot 7H_2O$, 0.05 g; $CaCO_3$, 0.02 g; $FeSO_4 \cdot 7H_2O$, 0.01 g; synthetic sea salt (Red Sea®), 36.0 g; deionized $H_2O$, 1.0 q.s./L. (2) *R5M* (per L): glucose, 10.0 g; yeast extract, 5.0 g; Casamino Acids, 0.1 g; $MgCl_2 \cdot 6H_2O$, 10.12 g; trace salts solution ($ZnCl_2$, 40.0 mg $L^{-1}$; $FeCl_3 \cdot 6H_2O$, 200.0 mg $L^{-1}$; $CuCl_2 \cdot 2H_2O$, 10.0 mg $L^{-1}$; $MnCl_2$, 10.0 mg $L^{-1}$; $Na_2B_4O_7 \cdot 10H_2O$, 10.0 mg $L^{-1}$; $(NH_4)_6Mo_7O_{24} \cdot 4H_2O$, 10.0 mg $L^{-1}$), 2.0 mL; synthetic sea salt (Red Sea®), 36.0 g; deionized $H_2O$, 1.0 q.s./L. (3) *ISP2* (per L): yeast extract, 4.0 g; malt extract, 10.0 g; dextrose, 4.0 g; synthetic sea salt (Red Sea®), 36.0 g; deionized $H_2O$, 1.0 q.s./L. (4) *Potato Dextrose*–PD (per L): potatoes infusion, 200.0 g; glucose, 20.0 g; synthetic sea salt (Red Sea®), 36.0 g; deionized $H_2O$, 1.0 q.s./L. (5) *A1* (per L): soluble starch, 10.0 g; yeast extract, 4.0 g; peptone, 2.0 g; synthetic sea salt (Red Sea®), 36.0 g; deionized $H_2O$, 1.0 q.s./L. (6) *PC-1* (per L) [28]: soluble starch, 10.0 g; peptone, 10.0 g; meat extract, 10.0 g; sugar cane molasses, 10.0 g; synthetic sea salt (Red Sea®), 36.0 g; deionized $H_2O$, 1.0 q.s./L. For all media the pH was adjusted between 7.8 and 8.2 using HCl 1.0 mol $L^{-1}$ or NaOH 1.0 mol $L^{-1}$.

## Extract preparation

After 1, 2, 3 and 7 days of cultivation, broths were centrifuged at 9,000 rpm for 10 min to separate the mycelium from the supernatant. Mycelium was extracted twice with 10 mL of MeOH/Acetone (1:1) for 10 min with a centrifugation step at 9,000 rpm for 10 min between each extraction, and then pooled together. Supernatant was extracted twice using 20 mL of EtOAc at 190 rpm for 30 min. Thus, two extracts were obtained per sample, one from the mycelium and one from the supernatant, *i.e.* the culture broth. Sterile culture media were used as controls (blanks of each culture media). After extraction, samples were dried in an Eppendorf® Concentrator Plus system under vacuum and the resulting material diluted to 1.0 mg $L^{-1}$ using MeOH HPLC grade.

## LC-MS/MS analyses

Liquid chromatography–mass spectrometry analyses were carried out on a HPLC (Shimadzu®) coupled to a mass spectrometer ESI-IT (Amazon SL, Bruker Daltonics), fitted with an electrospray ionization source and ion-trap analyzer. Chromatographic method consisted of solvent A (0.1% formic acid in $H_2O$) and B (0.1% formic acid in MeOH), starting at 5% up to 100% of B in 30 min followed by a hold of 100% of B for 5 min, using a C18 column (Phenomenex® Luna, 5 μm, 4.6 x 250 mm). The method employed a flow of 1.0 mL $min^{-1}$, 15 μL injection volume and column temperature of 40 ºC. Mass spectrometer was operated in positive mode, monitoring a mass range from 100 to 1500 atomic mass units (amu), capillary voltage of 3500 V, end plate offset of 500 V, nebulizer at 60 psi, dry gas 10 L $min^{-1}$ and dry temperature at 320˚C, using the untargeted mode (fragmentation at $MS^2$ level using a ramp of collision energy from 50 up to 75 eV).

## MS data processing

The LC-MS data was converted from.d to.mzXML using MSConvert software. Converted files were processed in three different ways: (i) for search in the spectral library and molecular networking from GNPS platform (gnps.ucsd.edu) for identification of known compounds and analogs [30]; (ii) using MZMine 2 software for qualitative chemical barcoding and (iii) to compare the relative amount of surugamides among samples.

## Molecular networking–GNPS

Converted mzXML files were uploaded to GNPS server (massive.ucsd.edu) using the cross-platform FTP client FileZilla. To construct the molecular network, the precursor and fragment ion mass tolerances were set at 0.8 and 0.2 Da respectively. Advanced network options were set as follows: minimum pairs cosine: 0.7; network TopK: 10; maximum connected component size: 100; maximum matched fragment ions: 4 and minimum cluster size: 2. For library search, a minimum of 4 matching peaks was chosen with a score threshold of 0.65. Default values were utilized for all other parameters. Construction of molecular network was performed using software Cytoscape, versions 2.8.2 and 3.6.1 [31]. Full molecular network colored by extracted broth/mycelium matrices and by culture media are in S1 and S2 Figs, respectively.

## MZmine 2 software processing and chemical barcoding

LC−MS profiles were also analyzed using chemical barcoding and principal component analysis as previously described [32]. All files were processed using MZmine 2 [33], where the converted files were submitted to the steps of mass detection, chromatogram building, deisotoping, alignment and exportation. Mass detection step generates a list of masses for each scan in the analysis. For this step, it was chosen a noise level value of $5.0 \times 10^5$ counts s$^{-1}$ by comparison with MeOH and media blanks [32]. Peaks presenting a lower intensity were therefore not detected and not included in the mass list for further analysis. The next step, chromatogram building, builds a chromatogram for each mass that can be detected in all scans from the mass list generated in the previous step. Then, deisotoping step was carried out for the selection of the representative ions in isotopic patterns. The last step in MZmine 2 software was the alignment mode, where $m/z$ and retention time ($r_t$) data were combined and all samples aligned in a single file. The alignment mode allows the conversion of a three-dimensional dataset to a two-dimensional dataset, where buckets are created with the peaks areas. For the peaks that are not present in the sample, a value of zero was assigned. Finally, the aligned peaks were exported as a.csv (comma separated value) file containing retention time, mass-to-charge ratio and peak area of all detected compounds in all samples. Relative quantification analysis of surugamides production over cultivation days and among media was performed using the peak areas of compounds retrieved in this step.

Standardization and artifact suppression were carried out in Microsoft® Excel 2010. First, presence-absence standardization was achieved transforming the dataset in a binary pattern, where peak areas greater than zero were given a value of one and peak areas equal to zero remained zero. For visualization of the chemical barcode, values of "1" within the dataset are shown as black ticks, while values of "0" are omitted. Ions detected in MeOH and media blanks were removed from samples for artifact suppression.

The chemical barcode is the full view of the table retrieved from MZMine 2, in which the main color groups distinguish the extracts obtained from either broth (orange tones) or mycelium (blue tones). Each colored strip within the color group indicates the extracts obtained after 1, 2, 3 or 7 days of growth; and each horizontal line of ticks within a strip represents a growth medium utilized for strain growth.

## PCA analysis

Statistical analysis was carried out using The Unscrambler (Camo software). Principal component analysis was performed for comparison and for identification of outliers with distinguishable chemical profile.

## Genomic analysis

The genomic DNA of BRB081 was extracted using the Wizard ® Genomic DNA Purification Kit (Promega Corporation, Fitchburg, USA), according to the manufacturer's recommendations, and the quality of the extraction was verified by agarose gel electrophoresis. The genomic DNA was sequenced by the Macrogen laboratory (South Korea, Seoul), using the HiSeq System platform (Illumina Inc., San Diego, USA). The Geneious R11 software (Biomatters, New Zealand) was used to assemble the reads into contigs, which were reordered based on synteny conservation by the MeDuSa server version 1.6 [34] using the complete genomes of *Streptomyces albidoflavus* J1074, *Streptomyces sampsonii* KJ40 and *Streptomyces koyangisis* VK-A60T as a reference. In both stages of assembly, data quality was assessed using the gVolante 1.2.1 software [35] with the choice of the BUSCO V1 pipeline [36]. Finally, the AntiSMASH 5.0 bacterial version software [37] was used to define biosynthetic genes clusters of secondary metabolites in the assembled genome of *Streptomyces* sp. BRB081 (GenBank accession number JACVQE010000000). The threshold of "rigor" parameter was set to "high", to present only well-defined clusters and containing all the necessary genes.

## Cytotoxicity evaluation–MTT assay

Cytotoxic activity was evaluated by the colorimetric MTT assay using a colon adenocarcinoma cell line (HCT-116 ATCC CCL-247). Cells were plated in a 96-wells plate at $0.6 \times 10^4$ cells/well ($3 \times 10^4$ cells/mL in 200 μL of medium). After 24h, crude extracts (10.0 mg mL$^{-1}$ in DMSO) were added in wells for a final concentration of 50 μg mL$^{-1}$. Plates were incubated for 72h and the supernatant substituted for culture medium containing MTT ((3-(4,5-dimethylthiazol-2-yl)-2,5-diphenyl tetrazolium bromide) at 0.5 mg mL$^{-1}$. After 3h, supernatant was removed, the plate dried and the precipitate containing formazan dissolved in 150 μL of DMSO. Finally, absorbance was measured at 570 nm [38]. The experiment was performed in duplicate. Doxorubicin and DMSO were used as positive and negative controls, respectively. Activities for each extract were expressed as percentage considering the positive control as 100% of inhibition. Extracts were considered cytotoxic when inhibited over 75% cell growth at the concentration tested.

## Results and discussion

GNPS is an open access web-based mass spectrometry platform that provides several tools to analyze MS data. The platform allows for annotation of compounds through spectral library search and organization of compounds (detected ions) into families of molecules with molecular networking [39]. In our work, we chose LC-MS/MS analysis and GNPS tools for data visualization and annotation of compounds through library search. Fig 1 shows the molecular network and the clusters with at least one library hit, along with their annotations.

Since GNPS platform provides annotation through comparison of fragmentation patterns, it is possible to suggest the structure not only of the compounds present in a sample, but also of analogues that show similar fragments in the mass spectrum. Therefore, new derivatives of known classes of compounds are detected and organized in clusters with other substances with similar structure.

Desferrioxiamine E (nocardamine) was annotated (Fig 1) along with four other desferrioxiamines (Desf, Desf-A1, Desf-D2 and Desf-03). These compounds are siderophores (iron chelating agents) commonly biosynthesized by actinobacteria through siderophore synthase superfamily pathway [40]. Furthermore, these compounds were mainly produced in PD medium. Studies suggest that this class of compounds has a role in bacterial growth and development [41] and may show antibacterial activity against mycobacteria [42]. A cluster of fatty

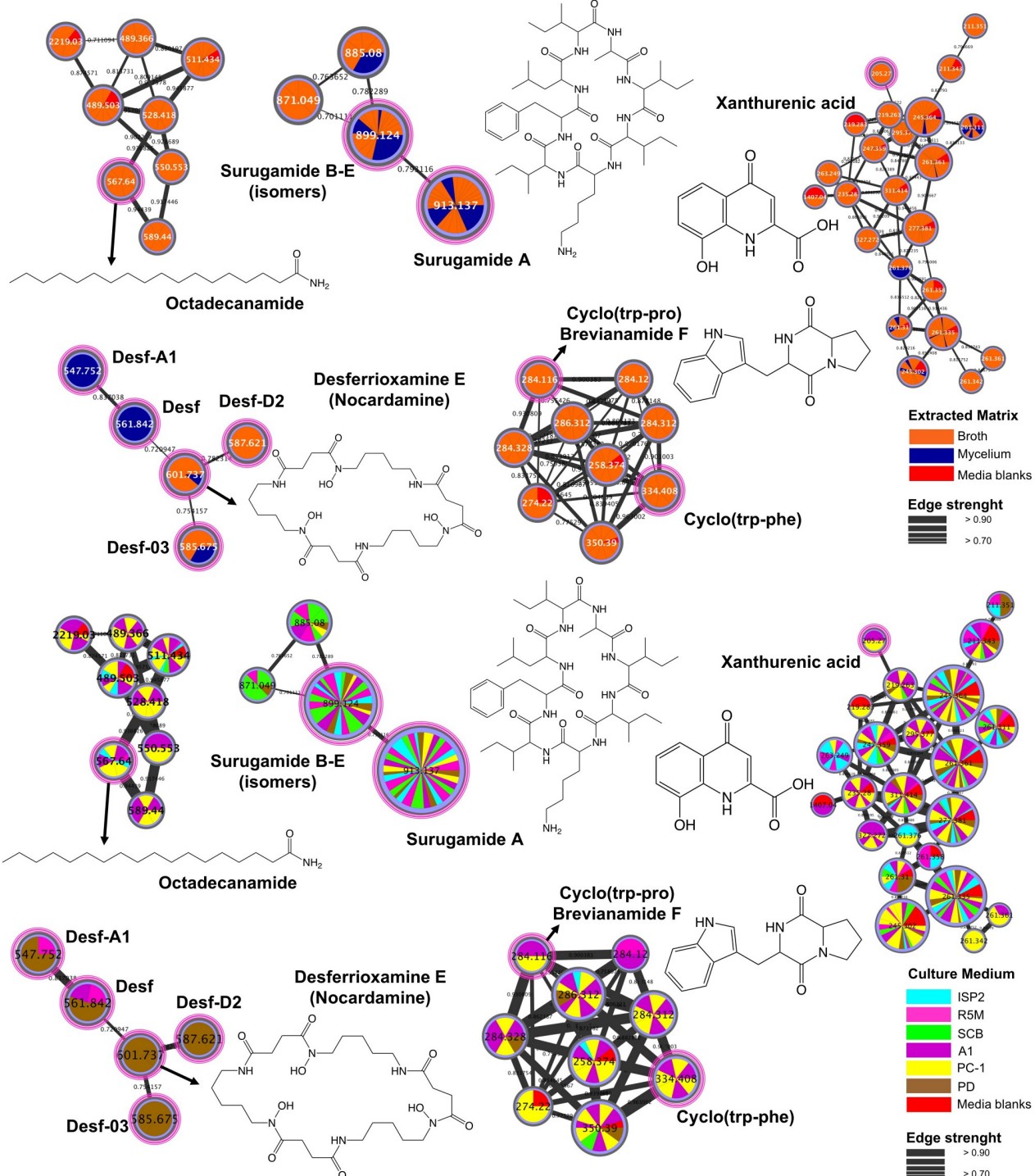

**Fig 1. Molecular network constructed with LC-MS/MS data obtained from the crude extracts produced by *Streptomyces* sp. BRB081 in six different media during seven days of cultivation.** Clusters of molecular families are depicted twice and show the detected compounds in broth and mycelium when both parts are separately extracted and across different media. Nodes corresponding to the spectral library match are colored in pink. The structures shown are related to annotated compounds.

amides was found containing octadecanamide (Fig 1). This compound was previously reported from a *Bacillus* sp. strain [43], although no report on bioactivity was found. Xanthurenic acid (XA), a metabolite produced both by Gram positive and negative bacteria, was also identified (Fig 1) [44]. XA is derived from tryptophan catabolism [45] and is considered an endogenous cell death factor which accelerates aging and disease development when accumulated in cells [46]. A cluster of diketopiperazines (Fig 1) allowed for annotation of the tryptophan-based diketopiperazines (DKPs) cyclo-(alanyl-tryptophanyl), cyclo-(phenylalanyl-tryptophanyl) and cyclo-(prolyl-tryptophanyl) (brevianamide F). DKPs are natural occurring nitrogen compounds produced by several marine microorganisms, including *Streptomyces* species [47]. These dimeric peptides present several biological activities, including antifouling [48], antiviral [49], antimicrobial and cytotoxic [50], among others. Tryptophan derivatives like the DKPs identified are especially interesting for presenting an indole moiety, which is common in medicinal chemistry and may influence several biological activities [51, 52]. Brevianamide F is also reported as a potential compound for treatment of cardiovascular dysfunction [53]. As shown, fatty amides, XA and DKPs were mainly produced in A1 and PC-1 media. An interesting cluster of the cyclic octapeptides surugamides was also detected (Fig 1), with the annotation of surugamide A and one or more isomers of surugamides B-E (all protonated analogues of *m/z* 899.1 were not distinguishable by mass spectrometry). It is possible to observe that, in general, they were produced in all media tested herein. These compounds were first isolated from a *Streptomyces* sp. strain and shown to be cathepsin B inhibitors [28]. Another cyclic octapeptide, champacyclin, has been reported as an isomer of surugamide A, differing in just two amino acid residues [54]. It is worth mentioning that several clusters found in the molecular network did not show any matches within databases, thus revealing a profusion of potentially unknown metabolites produced by this single strain.

Molecular networking provided a general profile of compounds produced by the *Streptomyces* sp. strain over time. In order to investigate if a compound is kept intracellular or secreted, samples from broth and mycelium extractions were colored in orange and blue, respectively, in the molecular network (Fig 2A). Furthermore, to identify which culture medium is better suited for the production of a specific metabolite, extracts obtained from BRB081 growth in each of the six culture media were also differently colored (Fig 2B). Fig 2 shows examples of two clusters, the surugamides (left), with two putatively new analogues, and another cluster, with no compounds identified (right). In both cases, it is possible to discern the matrix (broth/mycelium) that gives improved yields of these compounds (Fig 2A), as well as the best culture medium for their production (Fig 2B). Extraction of broth and mycelium independently usually provides clearer extracts. Thus, determining whether a target compound is only found in a specific part may guide towards a more efficient extraction and facilitate further isolation steps.

Fig 2A shows that the putative unknown compounds from the cluster in the right were found only in the broth, while the surugamides (cluster in the left), with the exception of ion of *m/z* 871.1, found only in the broth, were detected in extracts obtained from both matrices. Still, analyzing the extracted ion chromatograms for surugamides showed in Fig 3, at seven days of cultivation in SCB medium, these compounds presented a greater intensity and, consequently, higher concentration in the broth rather than in the mycelium extract. Despite published works reporting otherwise [28, 55], extraction of culture broth appears to raise the yield of surugamides.

This approach can be applied to compare compound production in different culture media. For example, the molecular network showed that the putatively new cluster on the right (Fig 2B) was only produced in SCB and PD media, but without quantitative information. Despite the interesting group of compounds, possibly further optimization of cultivation parameters will be necessary for monitoring and isolation of compounds prior scaling-up.

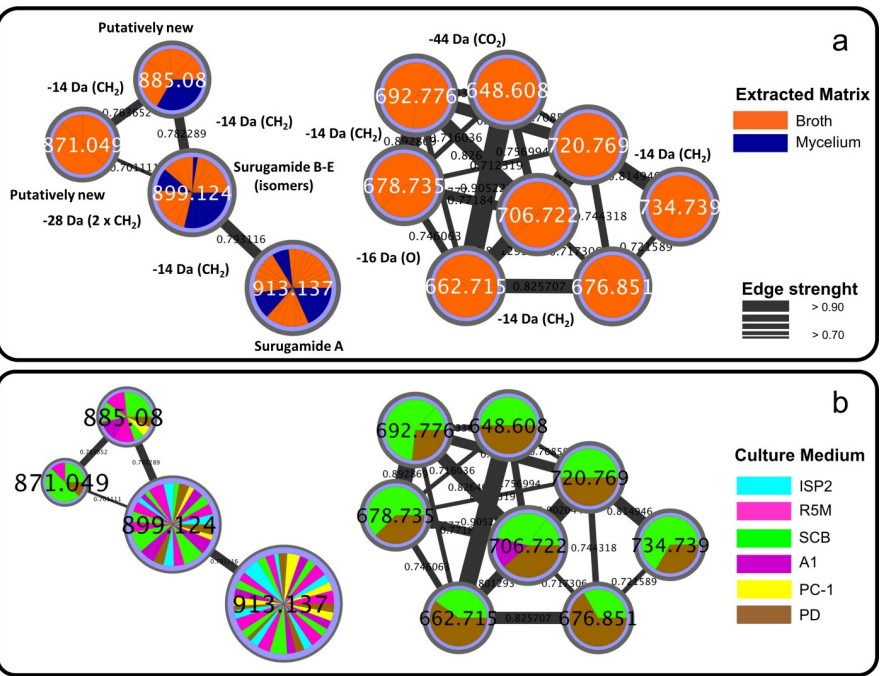

**Fig 2. Clusters from the molecular network of the identified octapeptides surugamides (left) and of putatively new unknown metabolites (right).** a) color-code discriminates nodes occurring in extracts obtained from the different matrix (broth/mycelium); b) color-code discriminates nodes occurring in extracts obtained from the different culture medium.

This shows the importance of checking the relative amount of compounds among samples, since molecular networks disregard peak areas. Integration of peaks provides a relative quantitation of a specific compound among samples and allows the selection of samples containing the highest concentration of a target metabolite. For example, surugamide A ($m/z$ 913.1, $[M+H]^+$) and the surugamide B-E isomers ($m/z$ 899.1, $[M+H]^+$) were detected in all media, while the putatively new surugamides of $m/z$ 885.1 ($[M+H]^+$) and $m/z$ 871.1 ($[M+H]^+$) were detected, respectively, in all media excepted ISP2 and only in SCB, R5M and PD. Since the mycelium was shown to be an inferior source of surugamides (Fig 3), only samples from broth extractions were considered for comparison of the relative amount of these compounds. Similarly, mycelium extractions from all media were poorer sources of surugamides. After plotting the peak areas for the surugamides detected in all media for each day of extraction, the maximum production of all analogues was achieved after seven days of cultivation in SCB medium (Fig 4). Notably, extracts obtained in SCB medium showed a much higher amount of

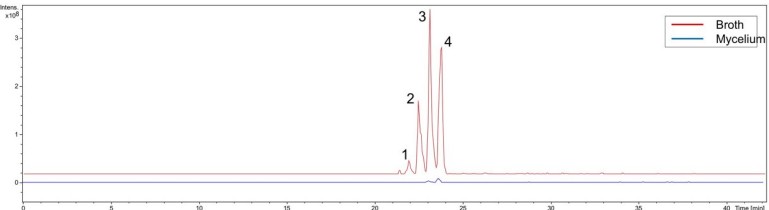

**Fig 3. Extracted-ion chromatogram from LC-MS analyses of broth (red) and mycelium (blue) extracts (SCB medium– 7 days).** Surugamides: 1 at 22.0 min, 871.1 Da; 2 at 22.4 min, 885.1 Da; 3 at 22.9 min, 899.1 Da; 4 at 23.5 min, 913.1 Da.

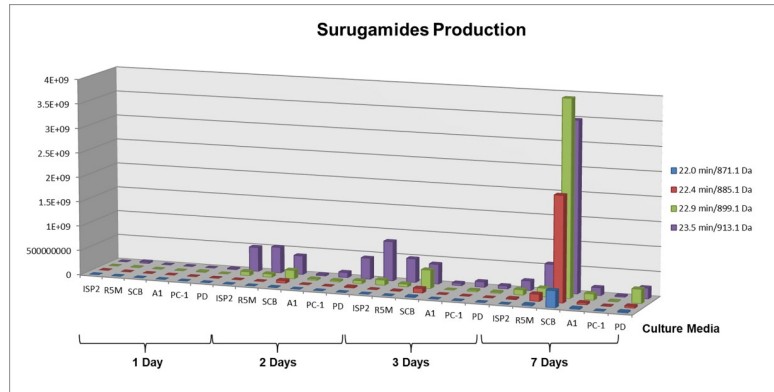

**Fig 4. Surugamides production over 1, 2, 3 and 7 days of cultivation.** Comparison of peak areas over time and culture media.

surugamides than those in PC-1, previously reported for surugamide production [28]. A recent study compared the production of surugamide A between two *Streptomyces* strains (J1074, recovered from a soil sample, and SM17, isolated from a marine sponge) using OSMAC and genome mining approach. Despite their sourcing from different environments, both strains belong to the same phylogroup and the biosynthetic gene cluster for surugamides (*sur* BGC) was found in their genomes. A search for *sur* BGC homologs in genomes available in the Genbank revealed that all five microorganisms that have this BGC are classified as *Streptomyces albidoflavus* and belong to the same phylogroup of J1074 and SM17 strains, suggesting the *sur* BGC might be exclusive to *S. albidoflavus* species. Moreover, a comparison of whole genome regions that contained *sur* BGCs of each isolate revealed that saline environment-derived *sur* BGCs likely share more genetic similarities amongst each other, rather than with those from terrestrial environments. The OSMAC approach surprisingly showed that the strain from the marine source produced higher yields of surugamide A in all tested conditions (with and without salt), reaching yields up to >13-fold higher in one of the tested media [56].

The production of surugamides by *Streptomyces* sp. BRB081 increased overtime, but this is not the case for all compounds detected during the seven days of cultivation. In this context, an alternative approach to understanding the time frame of compound production is the visualization of the chemical barcode (Fig 5), which is the fingerprint of all samples aligned by retention time and *m/z*, where each black tick represents a detected ion [32].

The visual representation of the chemical variance across different conditions in the chemical barcode allows an easy comparison of compound production regarding media composition, matrix extracted (broth/mycelium) and cultivation time. In Fig 5A, the upper arrow points to a group of compounds with a similar retention time of 25.33 min and *m/z* 348.4, *m/z* 514.9 and *m/z* 432.4 Da that were produced after 2 days of fermentation in PD medium and extracted from broth. However, after 7 days of cultivation in the same conditions, these ions are no longer seen (lower arrow), neither were these compounds observed in mycelium extracts from any media (blue region). In fact, this representation allows for a very straightforward comparison of broth and mycelium extracts. Regions 1 and 2 in Fig 5 correspond to compounds produced after 7 days of cultivation in all media detected in broth and mycelium extracts, respectively. It is easy to notice that the general profile of compounds extracted is very different and broth extractions provide much more diverse extracts when compared to mycelium.

Chemical barcoding is also useful to compare chemical diversity among the different growth media used. Fig 5B shows a comparison of the chemical profile of broth extraction of

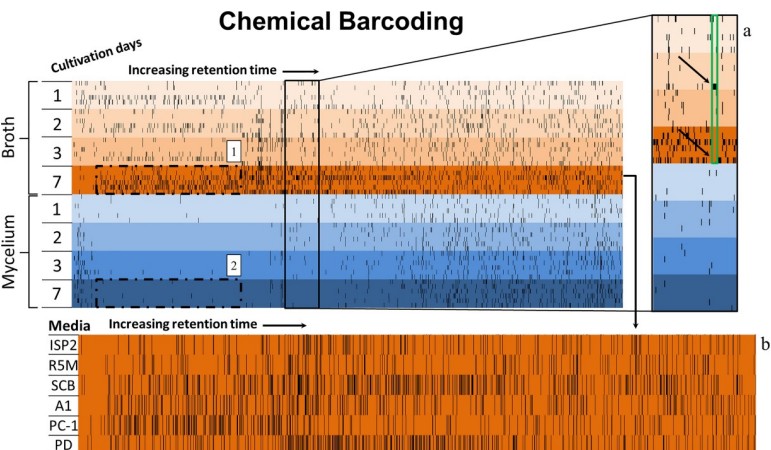

**Fig 5. Chemical barcode representing the diversity from broth and mycelium extracts of one single strain (*Streptomyces* sp. BRB081) cultivated in 6 different media between 1 and 7 days.** A) The detail shows compounds produced after 2 days of cultivation disappear after 7 days; B) enlargement of the fingerprint of extracts from broth after 7 days of cultivation, where each line shows vertical ticks representing ions detected in extracts obtained in each media used.

all media after 7 days of cultivation. Even with a few similarities, production of compounds by *Streptomyces* sp. BRB081 strain varied across all media. For example, PC-1 favored production of highly polar compounds (low retention time). Conversely, SCB led to compounds of medium and low polarity (medium and high retention time).

The table that originates chemical barcoding comprises thousands of ions, which can challenge the identification of samples with unique composition. PCA can be used as a tool to highlight outliers and indicate samples with interesting chemical profiles [32, 57]. The Scores plot from PCA evaluation (Fig 6) revealed that at 7 days of cultivation, the broth extracts from all media stood out from the rest of the samples. However, extracts from A1, PC-1, ISP2 and R5M media fell closer to zero, evidencing a lesser diversity of metabolites. At 7 days of cultivation in SCB and PD, the *Streptomyces* sp. strain produced unique compounds. This corroborates with the cluster represented on the right in Fig 2, where several putatively new compounds were produced only in these two media.

In Loadings plot (Fig 6) the distribution of ions is represented and the trend of compound production is observed. Highlighted groups A, B and C represent metabolites not common to most samples. Group A mainly comprises compounds that are shared by broth extract obtained in SCB medium at 7 days of cultivation and the group of samples from other media (7 days, broth), with the exception of that produced in PD media. Likewise, Group C includes compounds shared mainly by the broth extract from growth in PD medium at 7 days of cultivation and the group of samples from other media (7 days, broth) excepting SCB. Group B consists of compounds distributed through all media, but mostly in SCB and PD. The red circle indicates the ions corresponding to surugamides, which were, in fact, distributed across extracts produced in all media included in the cluster in Fig 2B, excluding the analogues of $m/z$ 871.1 (SCB, R5M, PD, PC-1 and A1) and 885.1 Da (SCB, R5M and PD). It is important to notice that these two analogues were mostly found in samples from SCB medium as indicated by the surugamides cluster separated by media (Fig 2B). The pink circle comprises the ions of $m/z$ 648.6, 678.7, 720.8, 734.7 and the other putatively new compounds from the cluster in Fig 2. As expected for compounds in Group B, they are mostly found in samples from SCB and PD media, confirming that the evaluation with PCA highlights outliers that indicate unique compounds in large sets of samples.

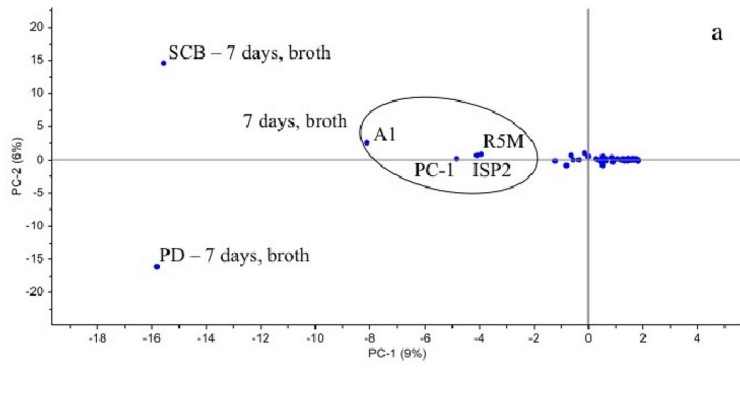

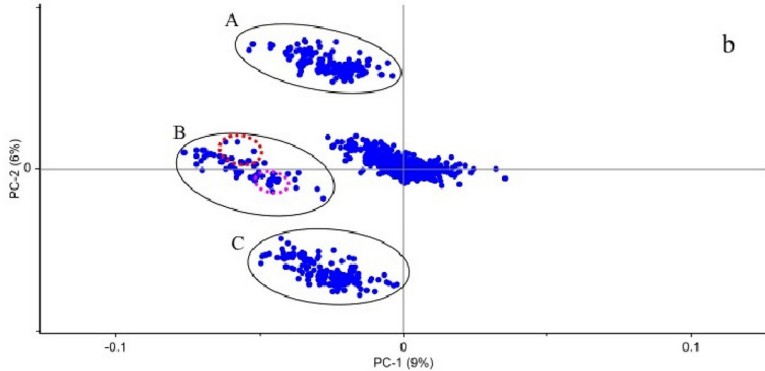

**Fig 6. PCA evaluation of all extracts obtained from cultivation of *Streptomyces* sp. strain BRB081.** a) Scores plot highlights samples with unique chemical composition. b) Loadings plot groups A, B and C combine the compounds responsible for the differences observed in Scores plot. The red region corresponds to surugamides analogues while the pink region comprises ions *m/z* 648.6, 678.7, 720.8, 734.7 and the other putatively new analogues from the cluster in Fig 2.

The genome analysis of BRB081 resulted in the prediction of four regions containing biosynthetic gene clusters (BGCs) of secondary metabolites that corroborate the metabolomic data. This approach allowed to correlate the compounds detected in the fermentation extracts with the genomic fractions dedicated to their biosynthesis (Table 1).

A BGC containing 100% of the gene composition related to the biosynthesis of siderophore desferrioxamine E by *Streptomyces* sp. ID38640 (MIBiG BGC0001478) was identified in region 1. Natural products containing diketopiperazine (DKP) can be biosynthesized by two systems: (I) non-ribosomal peptide synthetases (NRPSs) and (II) cyclodipeptide synthases (CDPs) [58]. We cogitate that NRPSs are responsible for the DKP biosynthesis in BRB081. The region 4 shares 10% of the gene content (OK006_RS36485, OK006_RS36480) described for the biosynthesis of pepticinnamin E (MIBiG BGC0002014), a secondary metabolite composed of a cinnamoyl moiety, a modified tripeptide nucleus and an ester-linked diketopiperazine unit. In

**Table 1. BGCs of secondary metabolites predicted for *Streptomyces* sp. BRB081.**

| Region | Type | Position | Similar MIBIG BGC | ID (%) |
|---|---|---|---|---|
| 1 | Siderophore | 2.238.291 to 2.248.819 | Desferrioxamin E | 100% |
| 2 | NRPS | 2.968.632 to 3.071.286 | Surugamides A/D | 85% |
| 3 | NRPS | 3.128.496 to 3.199.422 | SCO-2138 | 28% |
| 4 | NRPS | 3.223.017 to 3.270.715 | Pepticinnamin E | 10% |

addition, this region has 7% of the gene content (ALG65317.1) related to the biosynthesis of WS9326 (MIBiG BGC0001297), which has a cinnamoyl moiety identical to that of pepticinnamin E [59]. Analyzing the adjacencies, it was found an additional 5% of the genes (ALG65334.1, ALG65335.1) described for WS9326 biosynthesis in region 3, as well as an ORF encoding cytochrome p450, reported in most diketopiperazine BGCs catalyzing peptide pathways, and an ORF encoding a protein with the tryptophan 2,3-dioxigenase activity domain, found for example in the maremycin biosynthetic pathway [60]. These results indicate that the candidate cluster for DKP biosynthesis by BRB081 may be between regions 3 and 4. This is the only genomic evidence *in silico* (using this approach) for the biosynthesis of natural products containing diketopiperazine (DKP) by BRB081. However, the difficult correlation of DKPs with their producer BGC appears to be common [61, 62]. The surugamides biosynthetic gene cluster was identified in region 2, sharing 85% of the gene content related to the biosynthesis of these molecules by *Streptomyces albidoflavus* J1074 (MIBiG BGC0001792). In comparison to this similar BGC, two points can be highlighted: (1) the presence of a single ORF encoding an MFS transporter, found in two copies in BGC0001792, and the absence of an ORF encoding a transcriptional regulator of the TetR/AcrR family, found in a copy in BGC0001792; (2) the degree of identity found among ORFs of NRPS ranging from 68% to 86%, which may result in different domain specificities for these enzymes (S1 Table). This set of characteristics is compatible with the production of the unidentified types of surugamides revealed in the extract and we will undoubtedly dedicate more efforts to the experimental description of this BGC in the very near future.

Cytotoxicity evaluation of all crude extracts was carried out considering a 75% growth inhibition as parameter for positive activity (Fig 7A). Curiously, the most active sample (SCB—7 days, broth extract) was also the most different one as indicated by PCA evaluation, supporting a deeper investigation of its composition. Therefore, surugamides may be responsible for the

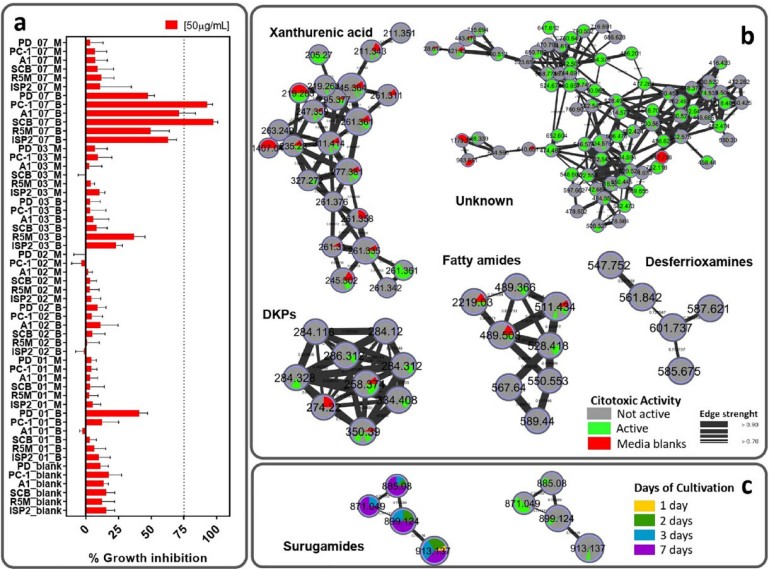

**Fig 7. Cytotoxic evaluation of the crude extracts (50 µg/mL) produced by *Streptomyces* sp. BRB081 on HCT-116 cell lines by the MTT assay.** a) Samples activities expressed as percentage considering the positive control as 100% of inhibition. Over 75% of growth inhibition threshold was determined as the parameter for positive activity. Samples code representation: Medium_Days_Part (B–Broth; M–Mycelium). b) Clusters of identified compounds colored by the activity they displayed (active/non active). c) Comparison between observed activity and days of cultivation of the surugamides cluster.

observed activity, since this sample gathers the highest content of this class of compounds (observed in Fig 4). Cyclic octapeptides may show cytotoxic activity as reported for champacyclin [54] and samoamide A, another cyclic octapeptide, which also presents cytotoxic activity against cancer cell lines [63]. However, PC-1 also showed a great cytotoxic activity, but not an expressive amount of surugamides. That means that the activity observed in this sample may accountable to other compounds produced mainly in this medium and not identified. Extracts obtained in these two media differed greatly in their chemical barcoding (Fig 5B), since PC-1 mainly showed metabolites of higher polarity than SCB. This could mean that the compounds responsible for the cytotoxicity in PC-1 extract may not be the same as those from SCB extract and worth further investigation. Insertion of cytotoxicity evaluation in the molecular network (Fig 7B) indicated which clusters are present in active samples and therefore may be responsible for activity. While the mere presence of a compound is not determinant for the activity of an extract–but also the concentration in which it occurs–, a large unknown cluster was highlighted in the network comprising several analogues in active samples. This cluster was produced in both SCB and PC-1 media and may also contribute to the observed activity. Besides bacteria, other marine microorganisms, such as endophytic fungi, are known to produce several types of cytotoxic compounds and have been sourced from several different environments. Some unique cytotoxic compounds, like the first methylthio-substituted aspochalasin derivative, active against prostate and colorectal cancer cell lines, was isolated from an *Aspergillus* strain recovered from the marine isopod *Ligia oceanica* [64]. Furthermore, the sponge-associated *Penicillium chrysogenum* produces the alkaloid sorbicillactone A, a potential active compound against leukemia cells. Another endophytic fungus from a marine alga, *Aspergillus nidulans* var. *acristatus*, was found to be the producer of arugosins G and H, which showed significant antitumor activity [65]. Marine fungi from *Talaromyces* genus and *Alternaria* sp. ZJ9-6B were found to produce, respectively, the mammalian DNA polymerase inhibitors kasanosins A and B, and alterporriol L, a cytotoxic bianthraquinone derivative [66]. However, the only clinically available metabolite derived from a marine fungus is plinabulin, currently undergoing phase III clinical trials for the treatment of non-small cell lung cancer and brain tumors. This compound is a cyclic dipeptide analog of halimide, a DKP isolated from *Aspergillus ustus* found in association with green algae *Halimeda* sp. [64].

Analyzing the surugamides cluster in Fig 7C, it can be noted that, despite the occurrence of surugamides, many samples were recognized as not active. This may be explained when considered the relative amount of surugamides present in the samples (Fig 4). Since mass spectrometry is a very sensitive technique, even compounds present in extremely low amounts can be detected and, therefore, shown in the molecular network. The lack of activity observed in other samples containing surugamides may be due to the low concentration of these compounds, since the active samples correspond to those produced after 7 days of cultivation. This shows the importance of combining techniques in metabolomics studies for prioritization of strains. Analyzing only the surugamides cluster disconnected from bioactivity could have deviated the attention from this interesting class of cyclic octapeptides.

## Conclusion

Metabolomics strategies and biological activity assays guide bioprospecting towards new bioactive compounds in natural products research. However, using only one strategy may be misleading since each approach has its own strengths and limitations. Combination of GNPS identification and molecular networking capabilities with MZmine 2 data treatment allowed the identification of compounds and evaluation of the best conditions for production among the parameters tested, thus expanding visualization of the compounds produced by

*Streptomyces* sp. strain BRB081 across several conditions. GNPS capabilities provided fast identification of compounds produced and indicated potential new analogues. MZmine 2 allowed further investigation calculating the relative amount of target compounds among samples, which correlated with the activity observed. Furthermore, genomic analysis provided consistency to metabolomic data indicating biosynthetic pathways. Cytotoxicity of the extracts could be attributed to higher yields of surugamides found in that obtained in SCB medium but is not justified in the extract obtained in PC-1, suggesting the presence of other interesting bioactive compounds produced by BRB081. The marine environment is an invaluable source of new bioactive metabolites. However, tools to amply explore such richness and unveil new chemical entities are still underdeveloped. The described approach not only revealed potentially new compounds, but, through chemical barcode, imaged a general profile and also the variations in the *Streptomyces* sp. strain metabolome as a whole.

## Supporting information

**S1 Fig. Molecular network constructed with LC-MS/MS data from the crude extracts produced by *Streptomyces* sp. BRB081.** Six different media were used during seven days of cultivation. Nodes colored by extracted matrix.
(TIF)

**S2 Fig. Molecular network constructed with LC-MS/MS data from the crude extracts produced by *Streptomyces* sp. BRB081.** Six different media were used during seven days of cultivation. Nodes colored by extracted cultivation media.
(TIF)

**S3 Fig. Picture of *Streptomyces* sp. strain BRB081 in A1 agar medium.**
(TIF)

**S4 Fig. Optical microscope image of *Streptomyces* sp. strain BRB081 with 40x magnification.**
(TIF)

**S1 Table. Annotated ORFs for the predicted region as a cluster of surugamides biosynthetic genes in *Streptomyces* sp. BRB081.**
(DOCX)

## Author Contributions

**Conceptualization:** Marcelo M. P. Tangerina, Marcelo J. Pena Ferreira.

**Formal analysis:** Marcelo M. P. Tangerina.

**Funding acquisition:** Norberto P. Lopes, Leticia V. Costa-Lotufo, Marcelo J. Pena Ferreira.

**Investigation:** Marcelo M. P. Tangerina, Luciana Costa Furtado, Vida M. B. Leite, Anelize Bauermeister, Karen Velasco-Alzate, Leandro M. Garrido.

**Methodology:** Marcelo M. P. Tangerina.

**Project administration:** Marcelo J. Pena Ferreira.

**Supervision:** Gabriel Padilla, Norberto P. Lopes, Leticia V. Costa-Lotufo, Marcelo J. Pena Ferreira.

**Writing – original draft:** Marcelo M. P. Tangerina, Luciana Costa Furtado, Vida M. B. Leite.

**Writing – review & editing:** Anelize Bauermeister, Paula C. Jimenez, Leandro M. Garrido, Gabriel Padilla, Norberto P. Lopes, Leticia V. Costa-Lotufo, Marcelo J. Pena Ferreira.

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
