## [Decision Letter · Decision Letter 0]

7 Aug 2020

PONE-D-20-13304

Metabolomic Study of Marine *Streptomyces* sp.: Secondary Metabolites and the Production of Potential Anticancer Compounds

PLOS ONE

Dear Dr. Marucci Pereira Tangerina,

Thank you for submitting your manuscript to PLOS ONE. After careful consideration, we feel that it has merit but does not fully meet PLOS ONE’s publication criteria as it currently stands. Therefore, we invite you to submit a revised version of the manuscript that addresses the points raised during the review process.

MS is now reviewed and we believe this MS needs significant corrections before we consider this MS for publication in PLOS One. Kindly do the needful changes as suggested by the reviewers and submit revised MS.

We look forward to receiving your revised manuscript.

Kind regards,

Vijai Gupta, PhD in Microbiology

Academic Editor

PLOS ONE

Journal Requirements:

Additional Editor Comments (if provided):

MS is now reviewed and we believe this MS needs significant corrections before we consider this MS for publication in PLOS One. Kindly do the needful changes as suggested by the reviewers and submit revised MS.

Reviewers' comments:

Reviewer's Responses to Questions

**Comments to the Author**

1. Is the manuscript technically sound, and do the data support the conclusions?

Reviewer #1: Yes

Reviewer #2: Yes

2. Has the statistical analysis been performed appropriately and rigorously? 

Reviewer #1: Yes

Reviewer #2: Yes

3. Have the authors made all data underlying the findings in their manuscript fully available?

Reviewer #1: Yes

Reviewer #2: Yes

4. Is the manuscript presented in an intelligible fashion and written in standard English?

Reviewer #1: Yes

Reviewer #2: Yes

5. Review Comments to the Author

Reviewer #1: It is a good work which needs to be published in Plos One after suggested revisions. The manuscript revolves around the drugs derived from the marine bacteria.

Variation in the expression of surugamides with increase in time-points needs more discussion.

In addition to bacteria, please include a line of discussion on about the other marine endophytic microorganisms which serve as a great source of cytotoxic compounds. This information has been thoroughly discussed in following articles which may be cited.

1. Uzma et al., Endophytic Fungi-Alternative Sources of Cytotoxic Compounds: A Review, PMID: 29755344, DOI: 10.3389/fphar.2018.00309.

2. Pandey A. (2019) Pharmacological Potential of Marine Microbes. In: Arora D., Sharma C., Jaglan S., Lichtfouse E. (eds) Pharmaceuticals from Microbes. Environmental Chemistry for a Sustainable World, vol 28. Springer, Cham

3. Bramhachari P.V., Anju S., Sheela G.M., Komaraiah T.R., Venkataiah P., Prathyusha A.M.V.N. (2019) Secondary Metabolites from Marine Endophytic Fungi: Emphasis on Recent Advances in Natural Product Research. In: Singh B. (eds) Advances in Endophytic Fungal Research. Fungal Biology. Springer, Cham

Inclusion of Discussion section is essential to provide a comprehensive view to the readers about the interpretation of results and literature.

Typographical errors must be rectified

Reviewer #2: Dear Authors,

The work represented is quite extensive and elaborate; which has made my work as a reviewer quite easy. The data mining efforts along with biological activity verification's is the best way forward in the quest for unique compounds. Marine actinobacteria is one of most current area of research and hence kudoos for exploring it.

Having said that there are a few minor queries and suggestions from my part regarding the work.

(a) Why don't you add a supplementary file with respect to the biochemical profile and morphological characteristics of Streptomyces sp. BRB081 along with its morphological image. I advise you to do so given the uniqueness of this strain and because its not been identified to the species level

(b) The 16S rRNA gene sequence and whole genome analysis data availability in GenBank NCBI with its accession no is missing. Kindly add them for authenticity. Without this the data mining is irrelevant and future comparison studies and uniqueness will be less explored.

(c) Cyto-Toxicity studies has been carried out on colon adenocarcinoma cell line (HCT-116 ATCC CCL-247). Is there a reason only this particular cell lines were chosen. Kindly justify?

(d) Surugamides is the particular secondary metabolite highlighted in this study. Kindly compare this compound availability, quantification and activity among other streptomyces genera, especially amongst the marine actinobacteria globally.

(e)7th day old Streptomyces sp. BRB081 extracts has been explored extensively in this study. Is there a basis why only 7th day of culture inocula should be the one to be explored. What happens to the secondary metabolite secretions on an advanced days of incubation?

I will emphasis deeply on query (b). Good Luck and try to answer to these queries with an open and scientific mind.

6. PLOS authors have the option to publish the peer review history of their article (what does this mean?). If published, this will include your full peer review and any attached files.

Reviewer #1: No

Reviewer #2: **Yes: **Vincent Vineeth Leo

---

## [Author Response · Author response to Decision Letter 0]

18 Sep 2020

Dear Dr. Gupta,

Academic Editor 

PLOS ONE

The manuscript is now revised and all suggestions provided by referees were implemented. These modifications are highlighted in the text with enabled “Track Changes” as required. Additionally, some specific questions from the reviewers were answered point by point below.

Additional requirements to address from the editor:

Question 1. Please ensure that your manuscript meets PLOS ONE's style requirements, including those for file naming. The PLOS ONE style templates can be found at 

The manuscript was revised and instructions provided in both templates were strictly applied to meet PLOS ONE’s style requirements.

Question 2. Please include captions for your Supporting Information files at the end of your manuscript, and update any in-text citations to match accordingly. Please see our Supporting Information guidelines for more information: http://journals.plos.org/plosone/s/supporting-information. 

Captions for Supporting information were included at the end of the manuscript.

Question 3. We note that you have included the phrase “data not shown” in your manuscript. Unfortunately, this does not meet our data sharing requirements. PLOS does not permit references to inaccessible data. We require that authors provide all relevant data within the paper, Supporting Information files, or in an acceptable, public repository. Please add a citation to support this phrase or upload the data that corresponds with these findings to a stable repository (such as Figshare or Dryad) and provide and URLs, DOIs, or accession numbers that may be used to access these data. Or, if the data are not a core part of the research being presented in your study, we ask that you remove the phrase that refers to these data. 

The phrase “data not shown” was removed since the data is not a core part of the research being presented.

Response to reviewer's questions:

Reviewer #1:

It is a good work which needs to be published in Plos One after suggested revisions. The manuscript revolves around the drugs derived from the marine bacteria.

Variation in the expression of surugamides with increase in time-points needs more discussion.

In addition to bacteria, please include a line of discussion on about the other marine endophytic microorganisms which serve as a great source of cytotoxic compounds. This information has been thoroughly discussed in following articles which may be cited. 

1. Uzma et al., Endophytic Fungi-Alternative Sources of Cytotoxic Compounds: A Review, PMID: 29755344, DOI: 10.3389/fphar .2018.00309.

2. Pandey A. (2019) Pharmacological Potential of Marine Microbes. In: Arora D., Sharma C., Jaglan S., Lichtfouse E. (eds) Pharmaceuticals from Microbes. Environmental Chemistry for a Sustainable World, vol 28. Springer, Cham

3. Bramhachari P.V., Anju S., Sheela G.M., Komaraiah T.R., Venkataiah P., Prathyusha A.M.V.N. (2019) Secondary Metabolites from Marine Endophytic Fungi: Emphasis on Recent Advances in Natural Product Research. In: Singh B. (eds) Advances in Endophytic Fungal Research. Fungal Biology. Springer, Cham

Inclusion of Discussion section is essential to provide a comprehensive view to the readers about the interpretation of results and literature.

Typographical errors must be rectified 

 Additional discussion about the expression of surugamides was added in lines 353-364 of the final manuscript. Also, suggested references were included and discussion of other marine endophytes was improved in lines 477-490. Combined Results and discussion section was chosen following the guidelines of PLOS ONE. The authors believe that showing the discussion immediately after the results provides a better understanding of the work done.

Reviewer #2:

The work represented is quite extensive and elaborate; which has made my work as a reviewer quite easy. The data mining efforts along with biological activity verification's is the best way forward in the quest for unique compounds. Marine actinobacteria is one of most current area of research and hence kudoos for exploring it. 

Having said that there are a few minor queries and suggestions from my part regarding the work. 

(a) Why don't you add a supplementary file with respect to the biochemical profile and morphological characteristics of Streptomyces sp. BRB081 along with its morphological image. I advise you to do so given the uniqueness of this strain and because its not been identified to the species level 

Pictures with the morphological characteristics of strain BRB081 were included in the Supporting Information (S3 and S4 Figs). Unfortunately, due to COVID-19 restrictions in our institution, the biochemical profile could not be done. However, the complete genome was deposited in the GenBank and can be found under the Accession No JACVQE000000000.

(b) The 16S rRNA gene sequence and whole genome analysis data availability in GenBank NCBI with its accession no is missing. Kindly add them for authenticity. Without this the data mining is irrelevant and future comparison studies and uniqueness will be less explored. 

The information was included in the manuscript (Streptomyces sp. BRB 081) Accession No JACVQE000000000.

(c) Cyto-Toxicity studies has been carried out on colon adenocarcinoma cell line (HCT-116 ATCC CCL-247). Is there a reason only this particular cell lines were chosen. Kindly justify? 

HCT-116 tumor cell line is one of the most used cell lines in screening programs along the world (Farnaes, La Clair & Fenical, Org. Biomol. Chem. 2014, 12(3), 418-423; Trzoss et al., PNAS 2014, 111(41), 14687-14692), and its use facilitate comparison with literature data. Besides, it has a moderate sensitivity to chemotherapeutic drugs, when compared to many solid tumor cell lines, generally allowing the selection of interesting extracts to pursue with chemical purification. 

(d) Surugamides is the particular secondary metabolite highlighted in this study. Kindly compare this compound availability, quantification and activity among other streptomyces genera, especially amongst the marine actinobacteria globally. 

 Additional discussion comparing the expression of surugamides among other marine actinobacteria was added in lines 353-364 of the final manuscript.

(e)7th day old Streptomyces sp. BRB081 extracts has been explored extensively in this study. Is there a basis why only 7th day of culture inocula should be the one to be explored. What happens to the secondary metabolite secretions on an advanced days of incubation? 

Cultivation for 7 days is a protocol of the lab and the interruption of that time can be done with fast-growing bacteria. The period of 7 days was selected because it has the highest ratio of colony yield (biomass) and extract yield without showing signs of culture decline and start of catabolism.

On behalf of all the authors, I am respectfully resubmitting the manuscript for your review. We sincerely hope that the changes we have made in response to the reviewer’s comments satisfy your expectations for publication.

Your sincerely,

Dr. Marcelo M. P. Tangerina 

marcelotangerina@usp.br

Institute of Biosciences 

University of São Paulo – USP 

São Paulo – SP, Brazil

---

## [Decision Letter · Decision Letter 1]

23 Nov 2020

PONE-D-20-13304R1

Metabolomic Study of Marine *Streptomyces* sp.: Secondary Metabolites and the Production of Potential Anticancer Compounds

PLOS ONE

Dear Dr. Marucci Pereira Tangerina,

Thank you for submitting your manuscript to PLOS ONE. After careful consideration, we feel that it has merit but does not fully meet PLOS ONE’s publication criteria as it currently stands. Therefore, we invite you to submit a revised version of the manuscript that addresses the points raised during the review process.

Manuscript "Metabolomic Study of Marine Streptomyces sp.: Secondary Metabolites and the Production of Potential Anticancer Compounds" needs minor revision before we may consider this MS for publication in PLOS One. Kindly do the needful corrections and submit a point-wise response. 

We look forward to receiving your revised manuscript.

Kind regards,

Vijai Gupta, PhD in Microbiology

Academic Editor

PLOS ONE

Additional Editor Comments (if provided):

Manuscript "Metabolomic Study of Marine Streptomyces sp.: Secondary Metabolites and the Production of Potential Anticancer Compounds" needs minor revision before we may consider this MS for publication in PLOS One. Kindly do the needful corrections and submit a point-wise response.

Reviewers' comments:

Reviewer's Responses to Questions

**Comments to the Author**

1. If the authors have adequately addressed your comments raised in a previous round of review and you feel that this manuscript is now acceptable for publication, you may indicate that here to bypass the “Comments to the Author” section, enter your conflict of interest statement in the “Confidential to Editor” section, and submit your "Accept" recommendation.

Reviewer #1: All comments have been addressed

Reviewer #2: All comments have been addressed

2. Is the manuscript technically sound, and do the data support the conclusions?

Reviewer #1: Yes

Reviewer #2: Yes

3. Has the statistical analysis been performed appropriately and rigorously? 

Reviewer #1: Yes

Reviewer #2: Yes

4. Have the authors made all data underlying the findings in their manuscript fully available?

Reviewer #1: Yes

Reviewer #2: Yes

5. Is the manuscript presented in an intelligible fashion and written in standard English?

Reviewer #1: Yes

Reviewer #2: Yes

6. Review Comments to the Author

Reviewer #1: The authors have adequately answered the questions raised by this reviewer. The manuscript can be accepted

Reviewer #2: Dear Authors,

The reviewer comments made were all addressed up-to certain standards. I understand that covid has prevented you from carrying out full fledged lab analysis. I have still found just a minor correction in the abstract prepared.

Here in abstract its written "extractions of mycelia and broth" (Line 24-25). But what kind of extraction is not specified. Is it solvent based secondary metabolite extraction?. Then it should be mentioned.

Also line 25 its written "crude extracts were analyzed"; what type of solvent based crude extract did u obtain? was it methanol extract or ethyl:acetone extract? Kindly specify, because such ambiguous statements are to be avoided.

Also one small concern the accession number provided in the manuscript "Streptomyces sp. BRB081 (GenBank accession number JACVQE000000000)" is not to be found in the NCBI GenBank database. Kindly re-verify this accession no. as this number provided might have been a temporary depository number.

Rectify these at the earliest.

Good luck

7. PLOS authors have the option to publish the peer review history of their article (what does this mean?). If published, this will include your full peer review and any attached files.

Reviewer #1: No

Reviewer #2: **Yes: **Vincent Vineeth Leo

---

## [Author Response · Author response to Decision Letter 1]

24 Nov 2020

Dear Dr. Gupta,

Academic Editor 

PLOS ONE

The manuscript was once again revised and all the minor changes suggested were implemented. These modifications are highlighted in the text with enabled “Track Changes” as required. Reviewer #1 stated that all questions were adequately answered. The specific questions from Reviewer #2 were answered below.

Question 1. Here in abstract its written "extractions of mycelia and broth" (Line 24-25). But what kind of extraction is not specified. Is it solvent based secondary metabolite extraction? Then it should be mentioned.

Also line 25 its written "crude extracts were analyzed"; what type of solvent based crude extract did u obtain? was it methanol extract or ethyl:acetone extract? Kindly specify, because such ambiguous statements are to be avoided.

The kind of extraction and the type of solvent based crude extract obtained were specified in the abstract as asked.

Question 2. Also one small concern the accession number provided in the manuscript "Streptomyces sp. BRB081 (GenBank accession number JACVQE000000000)" is not to be found in the NCBI GenBank database. Kindly re-verify this accession no. as this number provided might have been a temporary depository number.

This Whole Genome Shotgun project has been deposited at DDBJ/ENA/GenBank under the accession JACVQE000000000. The version described in this paper is version JACVQE010000000 and the number was replaced in the manuscript. The records will be available within a few days.

All modifications are highlighted in the text with enabled “Track Changes” as required. 

On behalf of all the authors, I am respectfully resubmitting the manuscript for your review. We sincerely hope that the changes we have made in response to the reviewer’s comments satisfy your expectations for publication.

 I also would like to disclose that we set aside a budget for the publication fees from a project that ends on the 11th of December, 2020. If possible, it would be very important for us if we can have a final decision before this deadline. Due to the financial crisis triggered by COVID-19 in Brazil, we face several cuts in science and our budget is currently very limited and restrict. 

Your sincerely,

Dr. Marcelo M. P. Tangerina 

marcelotangerina@usp.br

Institute of Biosciences 

University of São Paulo – USP 

São Paulo – SP, Brazil

---

## [Editor Report · Decision Letter 2]

9 Dec 2020

Metabolomic Study of Marine *Streptomyces* sp.: Secondary Metabolites and the Production of Potential Anticancer Compounds

PONE-D-20-13304R2

Dear Dr. Marucci Pereira Tangerina,

We’re pleased to inform you that your manuscript has been judged scientifically suitable for publication and will be formally accepted for publication once it meets all outstanding technical requirements.

Kind regards,

Vijai Gupta, PhD in Microbiology

Academic Editor

PLOS ONE

Additional Editor Comments (optional):

All the comments have been addressed.
---

## [Editor Report · Acceptance letter]

11 Dec 2020

PONE-D-20-13304R2 

Metabolomic Study of Marine *Streptomyces* sp.: Secondary Metabolites and the Production of Potential Anticancer Compounds 

Dear Dr. Tangerina:

I'm pleased to inform you that your manuscript has been deemed suitable for publication in PLOS ONE. Congratulations! Your manuscript is now with our production department. 

Kind regards, 

on behalf of

Dr. Vijai Gupta 

Academic Editor

PLOS ONE